https://doi.org/10.1038/s42003-021-01969-7　　**OPEN**
# Grb2 binding induces phosphorylation-independent activation of Shp2

Chi-Chuan Lin [1✉], Lukasz Wieteska[1], Kin Man Suen [1,2], Arnout P. Kalverda[1], Zamal Ahmed [3] & John E. Ladbury [1,4✉]

The regulation of phosphatase activity is fundamental to the control of intracellular signalling and in particular the tyrosine kinase-mediated mitogen-activated protein kinase (MAPK) pathway. Shp2 is a ubiquitously expressed protein tyrosine phosphatase and its kinase-induced hyperactivity is associated with many cancer types. In non-stimulated cells we find that binding of the adaptor protein Grb2, in its monomeric state, initiates Shp2 activity independent of phosphatase phosphorylation. Grb2 forms a bidentate interaction with both the N-terminal SH2 and the catalytic domains of Shp2, releasing the phosphatase from its auto-inhibited conformation. Grb2 typically exists as a dimer in the cytoplasm. However, its monomeric state prevails under basal conditions when it is expressed at low concentration, or when it is constitutively phosphorylated on a specific tyrosine residue (Y160). Thus, Grb2 can activate Shp2 and downstream signal transduction, in the absence of extracellular growth factor stimulation or kinase-activating mutations, in response to defined cellular conditions. Therefore, direct binding of Grb2 activates Shp2 phosphatase in the absence of receptor tyrosine kinase up-regulation.

[1] School of Molecular and Cellular Biology and Astbury Centre for Structural Molecular Biology, University of Leeds, Leeds, UK. [2] Wellcome Trust Cancer Research UK Gurdon Institute, University of Cambridge, Tennis Court Road, Cambridge, UK. [3] Department of Molecular and Cellular Oncology, University of Texas MD Anderson Cancer Center, Houston, TX, USA. [4] Department of Chemistry, Indian Institute of Technology Bombay, Mumbai, India. ✉email: C.C.Lin@leeds.ac.uk; j.e.ladbury@leeds.ac.uk

The reciprocal process of phosphorylation by kinases, and dephosphorylation by phosphatases, of selected residues regulates the intensity and longevity of intracellular tyrosine kinase-mediated signal transduction. The SH2 domain-containing tyrosine phosphatase 2, Shp2, (aka. PTPN11) plays a prominent role in this process in a multitude of receptor tyrosine kinase (RTK)-mediated signalling pathways, including activation of the extracellular signal-regulated kinase Erk1/2 (aka. mitogen-activated protein kinase, MAPK) pathway[1–5]. Shp2 is ubiquitously expressed in vertebrate cells and consists largely of, in sequential order: two Src homology 2 (SH2) domains (NSH2 and CSH2, respectively); a protein tyrosine phosphatase (PTP) domain; and a C-terminal tail with two tyrosyl phosphorylation sites (Y542 and Y580) and a proline-rich sequence (residues $^{559}$PLPPCTPTPP$^{568}$).

Shp2 was the first phosphatase to be identified as a human oncoprotein[6,7], and a large body of experimental and clinical studies has indicated that the hyperactivation of Shp2 contributes to tumour progression in, for example, breast cancer[8–11]. A great deal of interest has been shown in anticancer therapeutic approaches involving downregulation of Shp2[12–15]. However, despite extensive investigation, a lack of a mechanistic understanding of its upregulation in solid cancers has hindered pharmaceutical development. Existing inhibitors targeting phosphatase activity show low selectivity because of the highly conserved amino-acid sequences of phosphatase domains. However, more recently an approach based on small molecule stabilisation of the auto-inhibited conformation of Shp2 has shown efficacy[16].

Crystal structural detail revealed that Shp2 utilises an auto-inhibitory mechanism that prevails under basal conditions[17]. NSH2 forms an intramolecular interaction with the PTP domain, directly blocking access to the catalytic site, resulting in a 'closed' state. In this state, the NSH2 domain adopts a conformation that contorts its phosphopeptide binding cleft. Gain-of-function somatic mutations that result in the abrogation of the interaction between NSH2 and the PTP domain have been shown to be activating[7,18]. Auto-inhibition is released through one of two mechanisms involving the SH2 domains of Shp2. In the first, both N- and CSH2 interact with a binding partner including a phosphorylated bisphosphoryl tyrosine-based activation motif[19–23]. The second, more controversial, mechanism occurs under conditions where Shp2 has been phosphorylated, typically by an RTK[1,24,25]. This induces an intramolecular, bidentate interaction between the two phosphorylated tyrosine residues in the C-terminus of Shp2 and both NSH2 and CSH2[24,26,27].

We have previously investigated the constitutive control that the adaptor protein growth factor receptor-bound protein 2, Grb2, exerts over Shp2 in non-stimulated cells[28,29]. Grb2 consists of an SH2 domain sandwiched between two SH3 domains and is integral to several RTK-mediated signalling pathways. Non-phosphorylated Grb2 exists in a concentration-dependent dimer-monomer equilibrium ($K_d = 0.8\ \mu M$)[29]. Depletion of intracellular expression of Grb2 results in increased concentrations of monomer[30]. In addition, monomeric Grb2 (mGrb2) will also prevail when it is phosphorylated on tyrosine 160 (Y160) in the dimer interface. Under basal conditions, in the absence of growth factor stimulation, Grb2 cycles between the phosphorylated mGrb2, and the non-phosphorylated, typically dimeric state. The former is dependent on constitutive, background RTK activity, e.g., from fibroblast growth factor receptor 2 (FGFR2), whereas the latter results from concomitant Shp2 activity[30].

In this work, we provide molecular mechanistic detail on the activation of Shp2 in the absence of the two phosphorylation-dependent mechanisms highlighted above. Using the monomeric phosphorylation charge mimetic Grb2 mutant Y160E (Grb2$_{Y160E}$)[30], we show that in non-stimulated cells Grb2$_{Y160E}$ is able to greatly enhance Shp2 phosphatase activity via a bidentate interaction involving two discrete interfaces; (1) between the NSH2 domain of Shp2 and the SH2 of Grb2, and (2) between the PTP domain of Shp2 and the CSH3 of Grb2. The binding of Grb2$_{Y160E}$ releases Shp2 from its auto-inhibited state and results in an increase in the phosphatase activity independent of kinase-induced stimulation.

RTK-mediated signal transduction in cells that are not exposed to activating concentrations of extracellular stimuli is fundamental in maintaining homoeostasis and metabolic regulation. Aberrancies in this form of signalling can evoke cancer outcomes[31–33]. However, our appreciation of this form of signalling remains limited. This work provides a valuable example of how in the absence of growth factor stimulation a key enzyme can be upregulated and control MAPK pathway response.

## Results

**Shp2 interacts with monomeric Grb2 in the absence of growth factor.** Mutation of Y160 to glutamate (a phosphotyrosine charge mimetic) in the Grb2 dimer interface abrogates self-association of the adaptor protein[30,32]. To characterise the interaction between full-length Shp2 (Shp2$_{WT}$) and monomeric Grb2 (including Y160E mutation, Grb2$_{Y160E}$[25]) we initially used microscale thermophoresis (MST) to measure the affinity of the interaction of full-length proteins ($K_d = 0.33 \pm 0.04\ \mu M$; Fig. 1a, Table 1 and Supplementary Data 1). We demonstrated that Grb2$_{Y160E}$ can bind at two discrete sites on Shp2 using biolayer interferometry (BLI) on the following four GST-tagged phosphatase constructs: Shp2$_{WT}$, the tandem SH2 domains (residues 1–220: Shp2$_{2SH2}$), the PTP domain (251–524: Shp2$_{PTP}$) and a peptide corresponding to the C-terminal 69 amino acids (525–593: Shp2$_{C69}$). Both Shp2$_{WT}$ as well as Shp2$_{2SH2}$ polypeptides were able to interact with Grb2$_{Y160E}$. The truncated Shp2$_{PTP}$ bound to Grb2$_{Y160E}$ weakly, whereas the C-terminal tail failed to interact (Fig. 1b and Supplementary Data 1). These two binding sites were confirmed in an in vitro pull-down experiment in which Grb2$_{Y160E}$ was precipitated by both GST-Shp2$_{2SH2}$ and GST-Shp2$_{PTP}$ (Fig. 1c, Supplementary Fig. 4a and Supplementary Data 1). The interaction with GST-Shp2$_{PTP}$ was less pronounced. Interaction between Grb2$_{WT}$ and the Shp2 constructs appears to be negligible suggesting that, under the experimental conditions, the prevailing dimeric Grb2 is unable to interact. The limited complex formation seen with extended exposure of the blot is again presumed to be with the low population of monomeric protein at equilibrium (Fig. 1c inset, Supplementary Fig. 4a and Supplementary Data 1).

Upon phosphorylation, tyrosines 542 and 580 on the C-terminal tail of Shp2 are known binding sites for Grb2 SH2 domains[34]. To confirm our in vitro observation in a cellular context, we measured intracellular binding of Shp2$_{\Delta69}$ (deleted of 69 residues 525–593 to eliminate SH2-phosphotyrosine mediated interactions) to Grb2$_{Y160E}$. We used fluorescence lifetime imaging microscopy (FLIM) to detect stable complexes through fluorescence resonance energy transfer (FRET) between fluorophore-tagged proteins transfected into HEK293T cells under serum-starved conditions. FRET was confirmed by the left-shift of the fluorescence lifetime from the control lifetime measurements of donor (CFP-Grb2) in the presence of non-specific RFP-alone acceptor (2.2 nsec; Fig. 1d). No interaction between CFP-Grb2$_{WT}$ and RFP-Shp2$_{\Delta69}$ was observed since the fluorescent lifetime remains largely unaffected. Grb2$_{WT}$ exists in a monomer-dimer equilibrium and CFP-Grb2$_{WT}$ overexpression would favour dimer at the elevated adaptor protein concentration, hence limiting the availability of Grb2 to interact with Shp2$_{\Delta69}$. To circumvent concentration-dependent dimerisation, we used the monomeric CFP-Grb2$_{Y160E}$ in the FLIM-binding assay. FRET between CFP-Grb2$_{Y160E}$ and Shp2$_{\Delta69}$ shows a left-shifted fluorescence lifetime by 100 psec shorter than the RFP-alone control or

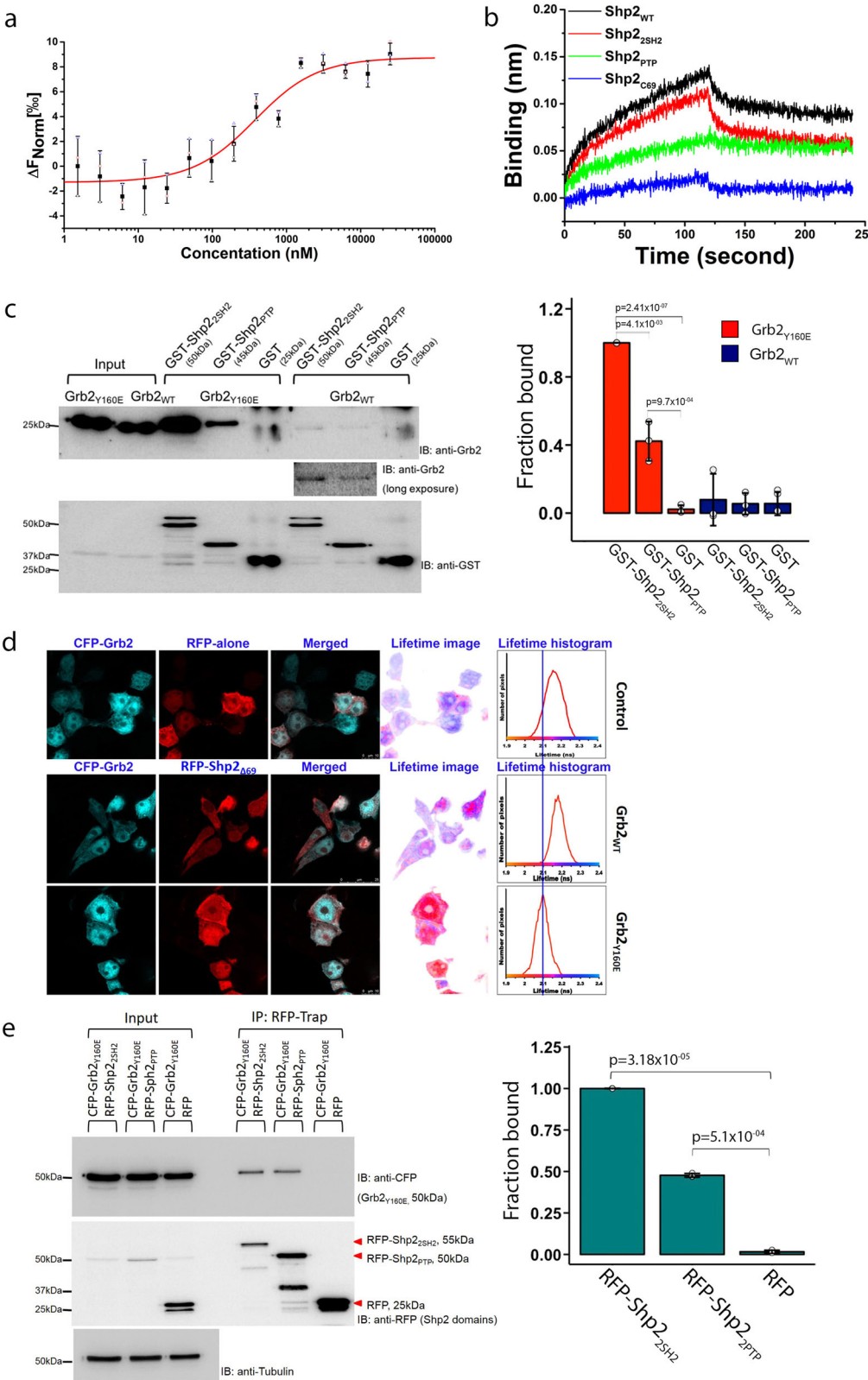

CFP-Grb2$_{WT}$ (Fig. 1d lower panel). This clearly demonstrates that only the monomeric Grb2$_{Y160E}$ interacts with Shp2. Immunoprecipitation further revealed that Grb2$_{Y160E}$ constitutively associates with Shp2$_{WT}$ in the absence of ligand stimulation (Supplementary Fig. 1). To further confirm the bidentate interaction in a cellular context, we co-expressed CFP-Grb2$_{Y160E}$ with RFP-Shp2$_{2SH2}$, RFP-Shp2$_{PTP}$, or

RFP alone in HEK293T cells. RFP-Trap immunoprecipitation demonstrates that both RFP-Shp2$_{2SH2}$ and RFP-Shp2$_{PTP}$ can precipitate CFP-Grb2$_{Y160E}$ (Fig. 1e, Supplementary Fig. 4b and Supplementary Data 1). Again, the interaction with GST-Shp2$_{PTP}$ was less pronounced. This result is consistent with our in vitro data shown in Figs. 1b and 1c.

**Fig. 1 Shp2 interacts with monomeric Grb2 in a phosphorylation-independent manner. a** MST measurement of full-length Shp2 binding to fluorescent labelled monomeric Grb2. Data are presented as (mean ± SD) of technical triplicates. For further details, see Table 1 and Methods. **b** BLI characterisation of individual Shp2 domains binding to GST-Grb2$_{Y160E}$ immobilised on a GST sensor. GST-Grb2$_{Y160E}$ was captured and the sensor and 10 µM of each Shp2 protein was used to test the interaction. Black: Shp2$_{WT}$, Red: Shp2$_{2SH2}$, Green: Shp2$_{PTP}$, Blue: Shp2$_{C69}$. The BLI sensorgram indicates that the Shp2$_{2SH2}$ and Shp2$_{PTP}$ mediate the interaction with Grb2. **c** GST pull-down experiment using GST-tagged Shp2$_{2SH2}$ and GST-tagged Shp2$_{PTP}$ to precipitate monomeric or dimeric Grb2 (Grb2$_{Y160E}$ or Grb2$_{WT}$, respectively). The pull-down results clearly demonstrate the interaction of Shp2 with monomeric Grb2. The blot represents three independent experiments. Inset: Densitometric analysis of GST pull-down results. Results are represented as mean ± SD. Statistics were determined using an unpaired Student's *t* test. **d** Fluorescence lifetime imaging microscopy (FLIM) displaying fluorescence resonance energy transfer (FRET) between CFP-Grb2$_{WT}$ and RFP-alone control (top); CFP-Grb2$_{WT}$ and RFP-Shp2$_{\Delta69}$ (middle); Grb2$_{Y160E}$ and RFP-Shp2$_{\Delta69}$ (bottom). The lifetime-image was generated using a false colour range pixel-by-pixel lifetime value corresponding to the average lifetime shown in the histograms. Two independent experiments were performed and each time 5–10 cells were counted. **e** RFP-trap immunoprecipitation of Shp2 domain complexed with CFP-Grb2$_{Y160E}$. This demonstrated that both Shp2$_{2SH2}$ and Shp2$_{PTP}$ domains can form complexes with Grb2$_{Y160E}$ in HEK293T cells. The blot represents three independent experiments. Inset: densitometric analysis of RFP-trap immunoprecipitation. Results are represented as mean ± SD. Statistics were determined using a paired Student's *t* test.

## Table 1 Binding affinities ($K_d$, µM) obtained from this study using MST.

| Target protein | Ligand | $K_d$ (µM) |
|---|---|---|
| Grb2$_{Y160E}$ | Shp2$_{WT}$ | 0.33 ± 0.04 |
| Grb2$_{Y160E}$ | Shp2$_{2SH2}$ | 0.28 ± 0.03 |
| Grb2$_{R86A/Y160E}$ | Shp2$_{2SH2\ R32A/R138A}$ | 0.15 ± 0.01 |
| Shp2$_{NSH2}$ | Grb2$_{SH2}$ | 33.8 ± 4.5 |
| Shp2$_{NSH2}$ | Grb2$_{NSH3}$ | 422 ± 21.5 |
| Shp2$_{NSH2}$ | Grb2$_{CSH3}$ | 207 ± 16.8 |
| Shp2$_{PTP}$ | Grb2$_{Y160E}$ | 0.30 ± 0.03 |
| Grb2$_{CSH3}$ | VLHDGD$^{297}$PNEP$^{300}$VSDYIN | N.B. |
| Grb2$_{CSH3}$ | TKCNNS$^{322}$KPKK$^{325}$SYIATQ | 173 ± 6.89 |
| Grb2$_{CSH3}$ | VERGKS$^{366}$KCVK$^{369}$YWPDEY | 122 ± 7.24 |
| Grb2$_{CSH3}$ | YGVMRV$^{386}$RNVK$^{389}$ESAAHD | N.B. |
| Grb2$_{CSH3}$ | AHDYTL$^{399}$RELKLSK$^{405}$VGQ | N.B. |
| Grb2$_{CSH3}$ | WPDHGV$^{429}$PSDP$^{432}$GGVLDF | 18.1 ± 0.188 |

Target proteins were fluorescently labelled. Serial dilutions of ligands were titrated in order to determine the binding affinity.

**The SH2 domain of Grb2 binds to the N-terminal SH2 domain of Shp2.** The affinity of Shp2$_{2SH2}$ binding to monomeric Grb2$_{Y160E}$ was determined by MST ($K_d = 0.28 ± 0.03$ µM; Fig. 2a, Table 1 and Supplementary Data 2). The possibility of binding of the pY-mimetic glutamate of Grb2$_{Y160E}$ to either of the SH2 domains of Shp2$_{2SH2}$ was ruled out because on mutating arginine residues normally required for pY binding in the respective SH2 domains of both proteins (on Shp2, R32A/R138A; Shp2$_{2SH2\ R32A/R138A}$ and on Grb2 R86A; Grb2$_{R86A/Y160E}$), Grb2$_{R86A/Y160E}$ was still capable of binding to Shp2$_{2SH2\ R32A/R138A}$ with similar affinity to the wild-type Shp2$_{2SH2}$ construct ($K_d = 0.15 ± 0.01$ µM; Fig. 2b, Table 1 and Supplementary Data 2). Having observed the Shp2$_{2SH2}$ interaction with Grb2, we sought to identify whether an individual domain was responsible for critical contact. GST-tagged Shp2$_{2SH2\ R32A/R138A}$, as well as the individual SH2 domains (N-terminal SH2 domain: Shp2NSH2 and C-terminal SH2 domain: Shp2CSH2), were used in pull-down experiments in which the NSH2, and not the CSH2, domain of Shp2 was shown to be sufficient for Grb2 binding (Fig. 2c, Supplementary Fig. 4c and Supplementary Data 2). Re-probing the blot with an anti-pY antibody revealed that the interaction was not mediated through phosphorylated tyrosine(s) or glutamate on Grb2$_{Y160E}$. This was further confirmed through in vitro binding assays using two phosphopeptides corresponding to the phosphorylatable tyrosine residues on Grb2 (pY160 and pY209), which show negligible interaction with Shp2$_{2SH2}$. (Supplementary Fig. 2a and Supplementary Data 5). Our data, therefore

are consistent with a non-canonical, phosphorylation-independent interaction between Shp2$_{2SH2}$ and Grb2$_{Y160E}$.

Having established the role of Shp2$_{NSH2}$ in binding to unphosphorylated Grb2$_{Y160E}$, we attempted to establish which domain(s) of Grb2 was required for recognition of the phosphatase. GST-tagged Shp2$_{NSH2}$ was captured on a GST BLI sensor and was exposed to the isolated SH2, as well as NSH3 and CSH3 domains of Grb2 (Grb2$_{SH2}$, Grb2$_{NSH3}$ and Grb2$_{CSH3}$, respectively). The data indicate that Grb2$_{SH2}$ binds to Shp2$_{NSH2}$ (Supplementary Fig. 2b), accompanied by a weaker interaction with Grb2$_{CSH3}$. MST experiments on the same isolated domains of Grb2 binding to Shp2$_{NSH2}$ confirmed that the interaction with Grb2$_{SH2}$ was dominant ($K_d = 33.8 ± 4.5$ µM). Interactions with both the SH3 domains of Grb2 were substantially weaker (Grb2$_{NSH3}$; $K_d = 422 ± 21.5$ µM and Grb2$_{CSH3}$; $K_d = 207 ± 16.8$ µM (Fig. 2d, Table 1 and Supplementary Data 2).

To assess the impact of the Grb2$_{SH2}$ domain binding to Shp2-NSH2 domain, we carried out NMR titration experiments based on 2D ($^1$H, $^{15}$N) HSQC spectra of $^{15}$N-Shp2-NSH2 domain in the absence, and the presence of an increasing concentration of unlabelled Grb2$_{SH2}$ domain. Peak assignments of Shp2$_{NSH2}$ backbone $^1$H-$^{15}$N resonances were derived from a suite of triple-resonance experiments (81% assignment coverage; Supplementary Fig. 3). The addition of Grb2$_{SH2}$ led to chemical shift perturbations (CSPs) for several resonances, indicating disruption of local chemical environments of specific amino acids as a result of complex formation (Fig. 2e). Comparison of the spectra of the free Shp2$_{NSH2}$, and the Shp2$_{NSH2}$-Grb2$_{SH2}$ complex showed that the average CSPs are relatively small. However, a limited number of residues show pronounced changes (Fig. 2f). The CSPs (>0.0075 ppm) were mapped onto a crystal structural representation of Shp2$_{NSH2}$ (PDB code: 2SHP; Fig. 2g, h). From this, it is possible to interpret a mechanism for Grb2$_{SH2}$-domain binding. The reported crystal structural data of Shp2 (PDB: 2SHP) reveal a non-phosphorylated auto-inhibited structure in which NSH2 directly interacts with the PTP domain, blocking the access of phosphorylated substrates. We see that a number of residues that are not within the Shp2-NSH2-PTP interface are perturbed by binding Grb2$_{SH2}$ (T12, V25, Q57 and Y100; Fig. 2h) and are likely to represent a discrete and extended interface formed between Grb2 and Shp2. In addition to this, we observe that a subset of residues that are within the intramolecular auto-inhibited interface also show CSPs (G39, F41, G60, Y62 and K70; Fig. 2f, g). For instance, G60 and Y62 are located on the DE loop, which inserts into the PTP domain catalytic site interacting with residues of the catalytic loop of Shp2[17]. This suggests that binding of Grb2$_{SH2}$ is also able to induce a distal conformational change in Shp2$_{NSH2}$ capable of disrupting the auto-inhibited intramolecular interface

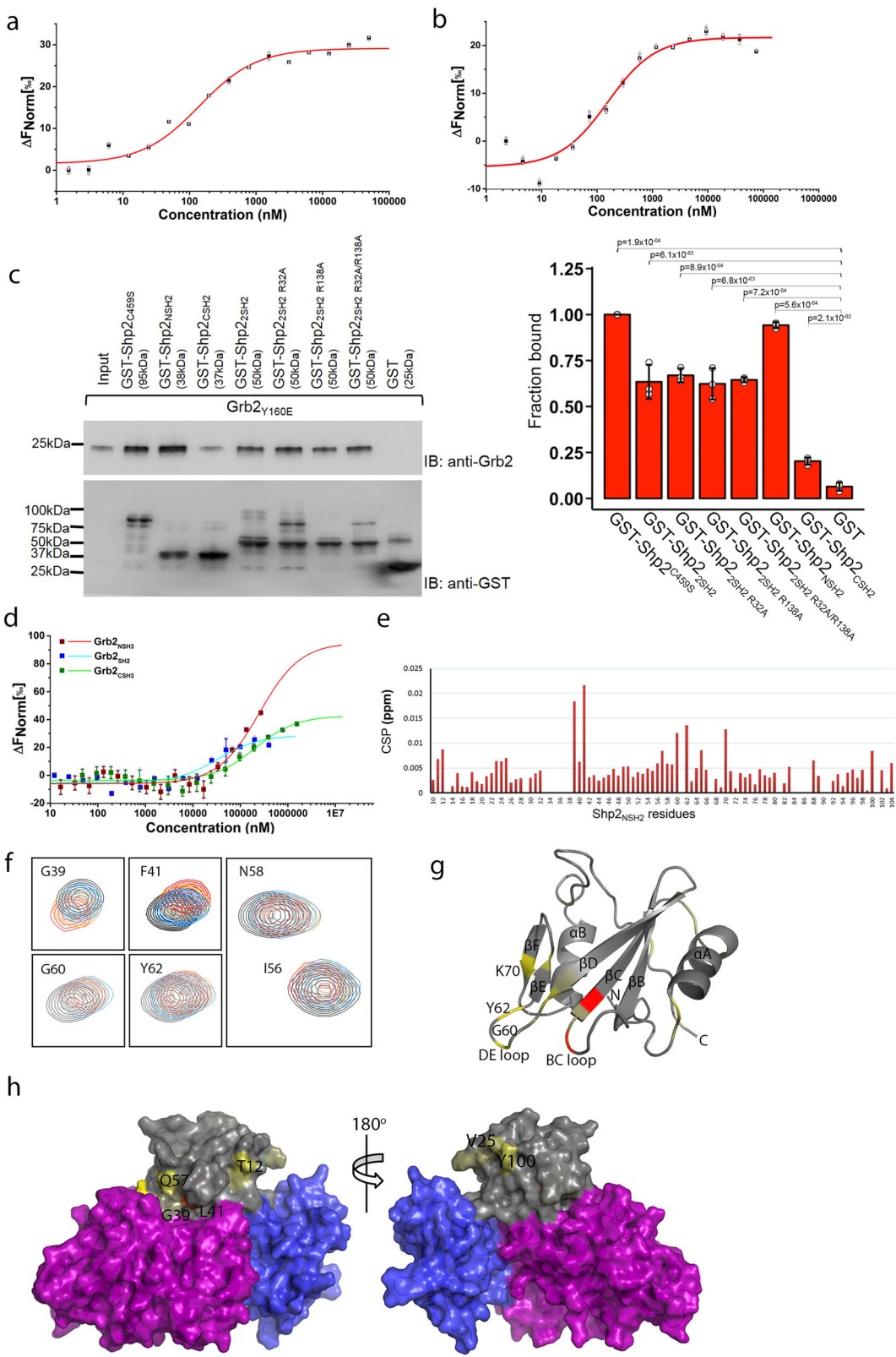

releasing the 'closed' structure and hence activating the phosphatase.

**The CSH3 domain of Grb2 interacts with Shp2$_{PTP}$.** Data shown in Figs. 1b and 1c revealed that, as well as binding to Shp2$_{2SH2}$, Grb2$_{Y160E}$ also formed an interaction with Shp2$_{PTP}$. We explored this further using MST and found that the isolated Shp2$_{PTP}$ binds Grb2$_{Y160E}$ with a similar affinity to the interaction with Shp2$_{2SH2}$ ($K_d = 0.30 \pm 0.03$ μM; Fig. 3a, Table 1 and Supplementary Data 3). To identify which domain of Grb2 interacts with Shp2$_{PTP}$ we used GST-fused Grb2$_{SH2}$, Grb2$_{NSH3}$ and Grb2$_{CSH3}$ constructs to precipitate recombinant Shp2$_{PTP}$. The pull-down indicated that only Grb2$_{CSH3}$ can bind to the PTP domain (Fig. 3b, Supplementary Fig. 4d and Supplementary Data 3). SH3

**Fig. 2 Identification of binding domain on Grb2 and Shp2$_{SH2}$ domains. a** MST measurement of Shp2$_{2SH2}$ binding to fluorescent labelled Grb2$_{Y160E}$. For further details, see Table 1 and Methods. Data are presented as (mean ± SD) of technical triplicates. **b** MST measurement of Shp2$_{2SH2\ R32A/R138A}$ binding to fluorescent labelled Grb2$_{R86A/Y160E}$. This experiment confirms a phosphorylation-independent interaction. Data are presented as (mean ± SD) of technical triplicates. **c** GST pull-down experiment using GST-tagged Shp2$_{NSH2}$, Shp2$_{CSH2}$, Shp2$_{2SH2}$ and its SH2 domain-deficient mutants. The pull-down results indicate that SH2 domain-deficient mutants have no effect on precipitating Grb2 and identify that the NSH2 of Shp2 mediates the interaction with Grb2. The blot represents three independent experiments. Inset: densitometric analysis of GST pull-down results. Results are represented as mean ± SD. Statistics were determined using a paired Student's $t$ test. **d** MST characterisation of Grb2 individual domains binding to fluorescent labelled Shp2$_{NSH2}$. The result indicates that the SH2 domain of Grb2 predominately binding to Shp2$_{NSH2}$. Data are presented as (mean ± SD) of technical replicates (SH2) or triplicates (NSH3 and CSH3). **e** Interaction of $^{15}$N-labelled Shp2$_{NSH2}$ and Grb2$_{SH2}$ determined by NMR chemical shift perturbation (CSP) plot for the interaction of Grb2$_{SH2}$ to $^{15}$N-labelled Shp2$_{NSH2}$. Minimal chemical shift perturbation upon Grb2$_{SH2}$ mapped on Shp2$_{NSH2}$ sequence. **f** Comparison of Shp2$_{NSH2}$ chemical shifts in the presence and absence of Grb2$_{SH2}$. Overlay of a region of the $^{15}$N-$^{1}$H HSQC (heteronuclear single-quantum coherence) spectra of Shp2$_{NSH2}$ and Grb2$_{SH2}$ complex (red) and of Shp2$_{NSH2}$ alone (black). **g** Mapping on the Shp2-NSH2 domain of the (PDB: 2SHP) of the consensus residues (painted light yellow to red) exhibiting strong CSPs upon Grb2$_{SH2}$ binding. **h** Mapping on the surface of Shp2 (PDB: 2SHP) of the consensus residues (painted light yellow to red) exhibiting strong CSPs upon Grb2$_{SH2}$ binding. The Shp2$_{NSH2}$ domain is in grey, Shp2$_{CSH2}$ domain is in blue and the SHp2$_{PTP}$ domain is in purple. This identifies the potential binding sites on Shp2$_{NSH2}$ for Grb2$_{SH2}$ binding.

domains recognise sequences usually incorporating the motif PXXP (where X is any amino acid), although the atypical R/KXXK motif has also been associated with SH3 domain recognition. We found that there are two PXXP motifs ($^{297}$PNEP$^{300}$ and $^{429}$PSDP$^{432}$) and four R/KXXK SH3 domain-binding motifs ($^{322}$KPKK$^{325}$, $^{366}$KCVK$^{369}$, $^{386}$RNVK$^{389}$ and $^{399}$RELKLSK$^{405}$) within the PTP domain, which could potentially serve as Grb2$_{CSH3}$ binding sites. We, therefore, performed MST assays using Grb2$_{Y160E}$ and six 16-residue peptides incorporating these sequences. Binding data showed that the Grb2$_{CSH3}$ had the highest affinity for the $^{429}$PSDP$^{432}$ on the Shp2$_{PTP}$ ($K_d \sim 18\ \mu M$; Fig. 3c, Table 1 and Supplementary Data 3). This interaction is ~60-fold weaker than the interaction between the intact PTP domain and Grb2$_{Y160E}$, suggesting that the peptides do not entirely represent the full site of interaction.

As Grb2 binds to Shp2 through two apparently discrete interactions largely represented by, (1) Grb2$_{SH2}$ binding to Shp2$_{NSH2}$, and (2) Grb2$_{CSH3}$ binding to Shp2$_{PTP}$, we used isothermal titration calorimetry (ITC) to qualitatively corroborate these two binding events. The binding isotherm shows a biphasic profile, resulting from an initial binding event with a stoichiometry of 2:1 Shp2$_{\Delta69}$:Grb2$_{Y160E}$, which was succeeded by a second binding event with a stoichiometry of unity as more Grb2$_{Y160E}$ was titrated (Fig. 3d and Supplementary Data 3). These data can be reconciled by a model in which in the initial injections Grb2$_{Y160E}$ is saturated by excess Shp2$_{\Delta69}$. Under these conditions, Shp2$_{\Delta69}$ will occupy both binding sites on Grb2 (i.e., Grb2$_{SH2}$ binding to Shp2$_{NSH2}$ on one Shp2$_{\Delta69}$ molecule, and Grb2$_{CSH3}$ binding to Shp2$_{PTP}$ on another Shp2$_{\Delta69}$ molecule). On further addition of Grb2$_{Y160E}$, based on the final stoichiometry of 1:1, the complex involves a bidentate interaction in which both the SH2 and CSH3 domain of a single Grb2 concomitantly bind to the NSH2 and PTP domains of a single Shp2, respectively.

**Grb2 increases activity independent of Shp2 phosphorylation.** The bidentate binding of Grb2, which includes an interaction with both the NSH2 and the PTP domains of Shp2 implies that complex formation could influence phosphatase activity. From our accumulated data, we speculate that the binding of monomeric Grb2 promotes a conformational change releasing Shp2 from the auto-inhibited state. To investigate this, we tagged Shp2$_{\Delta69}$ at both the N- and C-termini with blue fluorescent protein (BFP) and green fluorescent protein (GFP), respectively. When this pair of fluorophores are in close proximity (i.e., as would be expected in the auto-inhibited state), excitation of BFP results in fluorescence resonance energy transfer between the two fluorophores, and hence a reduction in the emission intensity from BFP with a concomitant increase from GFP. Conversely,

distancing the fluorophores through conformational change reverses this outcome. Steady-state FRET was measured whilst adding Grb2$_{Y160E}$ to Shp2$_{\Delta69}$ (Fig. 4a and Supplementary Data 4). The data clearly show that, prior to Grb2 addition, the BFP donor emission is low and GFP acceptor emission is high. Recovery of FRET donor BFP emission and a decrease in FRET acceptor GFP emission with increasing concentration of Grb2$_{Y160E}$ results in the growing population of Grb2$_{Y160E}$–Shp2$_{\Delta69}$ complex undergoing a conformational change from the auto-inhibited state to a state where the termini of Shp2$_{\Delta69}$ are separated.

To assess whether the Grb2 binding-induced conformational change affected enzymatic activity, we conducted an in vitro phosphatase assay. A pY-containing peptide substrate (END-pYINASL) was incubated with non-phosphorylated Shp2$_{\Delta69}$ and different concentrations of Grb2$_{Y160E}$. Free phosphate generated from hydrolysis of the pY was measured by the absorbance of a malachite green molybdate phosphate complex (Fig. 4b and Supplementary Data 4). A significant increase in phosphatase activity was measured with the increasing presence of Grb2$_{Y160E}$. Importantly, a similar level of turnover of the peptide was observed on comparing the monomeric Grb2-induced phosphatase activity to that observed for phosphorylated Shp2 in the absence of the adaptor protein (pShp2; Fig. 4c and Supplementary Data 4). These in vitro experiments suggest that under basal conditions Shp2 activity can be upregulated through binding to monomeric Grb2 alone, and that this activity is at least as high as occurs when the phosphatase is phosphorylated as seen in growth factor-stimulated cells.

**Monomeric Grb2$_{Y160E}$ is associated with the upregulation of Shp2 in cancer.** Shp2 activity has been shown to play a key role in cancer progression. As Grb2$_{Y160E}$ can promote phosphatase activity in the absence of Shp2 phosphorylation, this could lead to a proliferative outcome in cells through upregulation of the Erk1/2 pathway without the need for the elevated kinase activity often associated with cancer phenotypes. The role of upregulation of non-phosphorylated Shp2 through binding to monomeric Grb2 in cancer has not previously been investigated. For example, in the triple-negative breast cancer cell line MDA-MB-468 in the absence of extracellular stimulation, we see negligible phosphorylated Shp2 (pShp2). However, there is a significant concentration of phosphorylated Erk1/2 (pErk1/2; Fig. 4d, Supplementary Fig. 4e and Supplementary Data 4). These cells have a background level of monomeric phosphorylated Grb2. Indeed, Grb2 phosphorylation under basal non-stimulatory conditions has been reported for a number of cell lines[28] (Fig. 4d lower blot). These data are consistent with monomeric Grb2 being able to upregulate MAPK signalling (as shown by

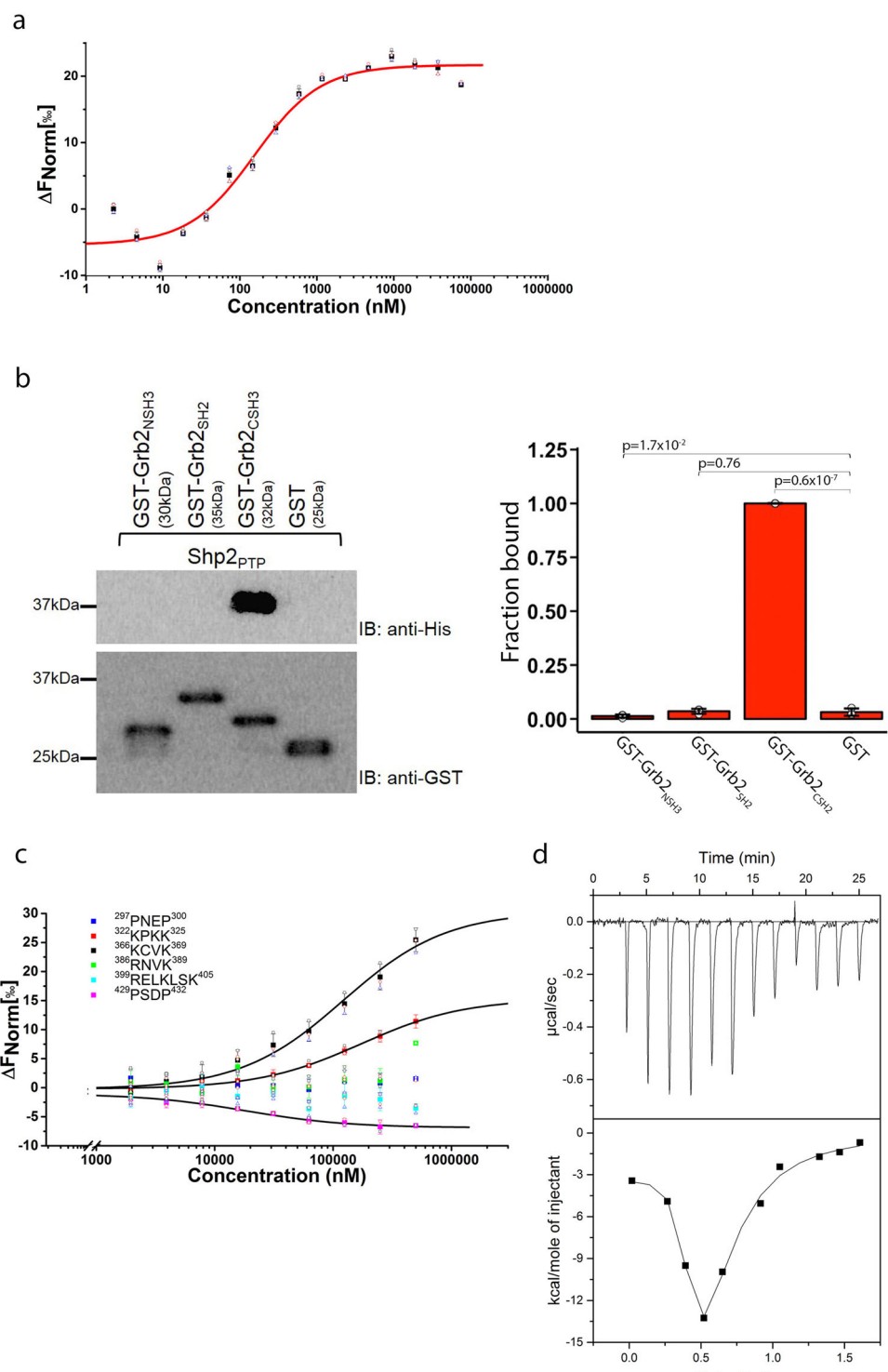

**Fig. 3 Characterisation of Grb2$_{Y160E}$ binding to Shp2$_{PTP}$ domain. a** MST measurement of Shp2$_{PTP}$ binding to fluorescent labelled Grb2$_{Y160E}$ (0.1 μM). Both PTP domain and tandem 2SH2 domains show similar affinity to Grb2$_{Y160E}$. Data are presented as (mean ± SD) of technical triplicates. **b** GST pull-down experiment using GST-tagged individual Grb2 domain (NSH3, SH2 and CSH3) to precipitate Shp2 PTP domain. The result clearly indicates the interaction is mediated through Grb2 CSH3 domain. The blot represents three independent experiments. Inset: densitometric analysis of GST pull-down results. Results are represented as mean ± SD. Statistics were determined using an unpaired Student's *t* test. **c** Sequence analysis identifies six potential PxxP or R/KxxK (x is any amino residue) motifs for Grb2 CSH3 domain interaction. Synthesised peptides were used to test the interactions with Grb2 CSH3 domain (Atto 488 labelled, 0.1 μM) using MST. The results show $^{322}$KPKK$^{325}$, $^{366}$KCVK$^{369}$ and $^{429}$PSDP$^{432}$ interact with Shp2 PTP domain. Motif $^{429}$PSDP$^{432}$ binds to Shp2 PTP domain with the highest affinity of 18 μM. See Table for detailed information and peptide sequences. Data are presented as (mean ± SD) of technical replicates (peptide 322–325 and peptide 429–432) or triplicates. **d** ITC was used to corroborate the two binding events between Shp2 and Grb2$_{Y160E}$. Grb2$_{Y160E}$ (200 μM) was titrated into Shp2$_{Δ69}$ (25 μM). Twelve 3 μl injections of Grb2$_{Y160E}$ were titrated into Shp2$_{Δ69}$. Top panel: baseline-corrected power versus time plot for the titration. Bottom panel: the integrated heats and the molar ratio of the Grb2$_{Y160E}$ to Shp2$_{Δ69}$.

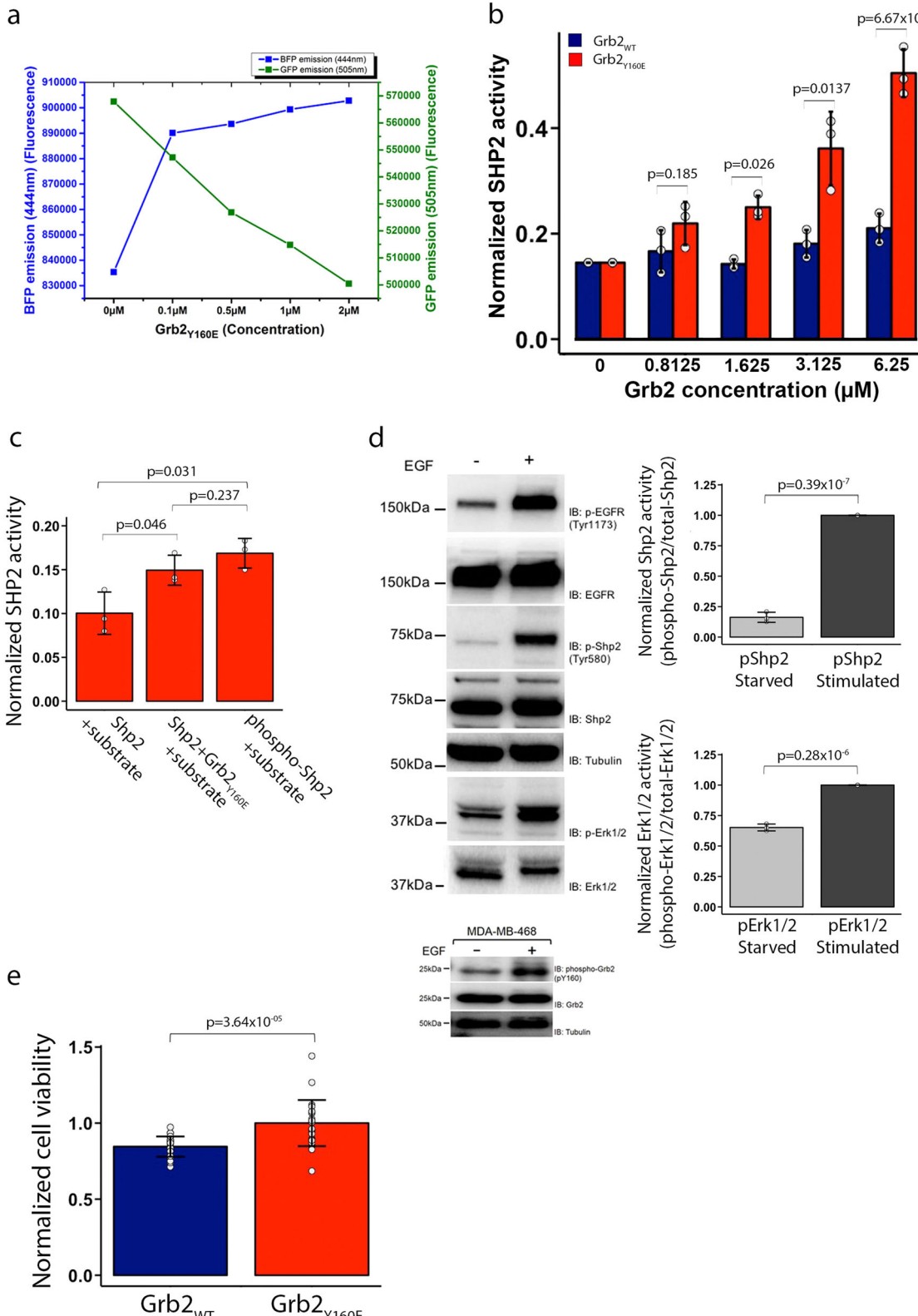

pErk1/2) through binding to Shp2 without the need for phosphorylation of the latter. Cell proliferation is a well-known marker for MAPK signalling upregulation so we investigated whether this was differentially affected by the presence of $Grb2_{WT}$ or $Grb2_{Y160E}$ expression using MDA468 cells. In the absence of extracellular stimulation, we observed that the expression of $Grb2_{Y160E}$ enhanced the level of cellular proliferation compared with wild type (Fig. 4e and Supplementary Data 4).

## Discussion

Kinase and phosphatase activity have to be precisely controlled to limit aberrant signal transduction in cells[35,36]. It has previously

**Fig. 4 Grb2$_{Y160E}$ upregulates Shp2 activity in a phosphorylation-independent manner. a** The effect of mGrb2 binding on Shp2 conformation was studied using steady-state FRET. 0.1 μM of Shp2$_{\Delta69}$ was N-terminally tagged with a BFP and C-terminally tagged with a GFP. Upon Grb2$_{Y160E}$ titration, the emission of both FRET donor (BFP) and acceptor (GFP) was recorded. **b** In vitro phosphatase assay using recombinant-unphosphorylated Shp2$_{\Delta69}$ (0.1 μM) and Grb2 (WT or Y160E). Free phosphate generated from hydrolysis of the pY from the substrate peptide (ENDpYINASL) was measured by the absorbance of a malachite green molybdate phosphate complex. The increasing concentration of Grb2$_{Y160E}$ gradually enhances Shp2 activity while the dimeric Grb2 has no effect on Shp2 activity. The graph represents the results of three independent experiments. **c** In vitro phosphatase assay using both recombinant-unphosphorylated (0.1 μM) and phosphorylated (0.1 μM) full-length Shp2 to compare the degree of enhanced phosphatase activity mediated by mGrb2 (10 μM) binding. This assay demonstrates that the mGrb2 binding-induced enhancement of Shp2 activity is comparable to the phosphorylated Shp2. The graph represents the results from two independent experiments, each independent experiment contained three technical repeats. **d** Activation of Shp2 and Erk1/2 in MDA-MD 468 cells in the absence of EGF ligand stimulation. The activation levels of both Shp2 and Erk1/2 in the absence (−) or presence (+) of EGF ligand stimulation are analysed and shown as densitometric analysis. The blot represents three independent experiments. Inset: Grb2 is tyrosine phosphorylated in MDA-MD-468 cells in the absence of EGF stimulation. **e** MTT cell proliferation assay of MDA-MD-468 cells expressing either Grb2$_{WT}$ (blue) or Grb2$_{Y160E}$ (red). The graph represents the results from two independent experiments, each independent experiment contained 24 technical repeats. Results are represented as mean ± SD. Statistics were determined using an unpaired Student's $t$ test.

been demonstrated that Shp2 function is dependent on RTK activity[37–41], and is accompanied by phosphorylation of tyrosine residues Y542 and Y580. This appears to facilitate a conformational change that abrogates an intramolecular interface between the NSH2 and PTP domains of Shp2 and relieves the auto-inhibited state[26]. Activated Shp2 can play a role in proliferative signalling through the Erk1/2 pathway[3,4,37].

In this work, we show that the upregulation of Shp2 can also be accomplished in the absence of RTK activity, through the binding of the monomeric adaptor protein Grb2 (mGrb2). Monomeric Grb2 will be present in cells in which the adaptor protein is expressed at low concentrations (dimer-monomer equilibrium $K_d = 0.8$ μM[29,30]), or as the result of phosphorylation on Y160 by constitutive, background RTK activity in the absence of ligand stimulation[28]. The SH2 and CSH3 domains of Grb2 form a bidentate interaction with the NSH2 and PTP domains of Shp2, respectively. Interaction with Shp2 in this way results in an mGrb2-dependent conformational change which opens the auto-inhibited state to facilitate phosphatase activity.

The relevance of a mechanism for Shp2 upregulation in non-stimulated cells is likely to be associated with the requirement of cells to maintain homoeostatic and metabolic 'house-keeping' function in the absence of extracellular stimulation. We have previously shown that similar functional activation of enzymes and adaptors associated with RTK-mediated signalling pathways occurs under basal conditions without the need for kinase upregulation (e.g., Plcγ1;[31] Shc[33]). This second tier of signalling (tier 2 signalling), below the profound and defined effects of full kinase activation resulting from extracellular stimulation, is likely to be important in maintaining cell viability and in responding to stress. Aberrancies in tier 2 signalling can also be associated with cancer pathology[31]. This appears to be reflected here where, through interaction with mGrb2, Shp2 can be engaged in signalling in cancer cells without the need for mutated, dysfunctional kinases. Therefore, in cells depleted for a given RTK, as is found in triple-negative breast cancer (e.g., in MDA-MB-468 cells, which do not express Her2; Fig. 4d), proliferative or metastatic signalling could be driven by the uncontrolled impact of monomeric Grb2-mediated Shp2 activation. The importance of the interaction with Grb2 may also be emphasised by the observation of two reported oncogenic Shp2 mutations in the NSH2, which are localised at the identified Grb2-binding site, namely T42A[42] and N58K[43]. Change-of-function associated with these mutations would potentially release regulatory control of background tyrosine kinase-mediated signalling.

Non-stimulatory conditions portray many features of the early phases of tumour development and progression as well as prevailing in resistant cells treated with kinase inhibitors. Thus, the understanding of how signalling proteins behave and are controlled under these conditions is the necessary foundation to develop more efficient therapeutic strategies. It remains to be seen whether examples of mGrb2-mediated activation of Shp2 are common regulators of signalling in basal cells, and particularly whether aberrancies resulting from environmental change-related stress can drive tumorigenesis through perturbation of Grb2 concentration.

## Methods

**Cell culture.** HEK293T cells were maintained in Dulbecco's modified Eagle's high glucose medium supplemented with 10% (v/v) fetal bovine serum and 1% anti-biotic/antimycotic (Lonza) in a humidified incubator with 10% $CO_2$.

**Reagents.** Recombinant human epidermal growth factor (EGF) was purchased from R&D Systems. Anti-Grb2 (sc-8034) were purchased from Santa Cruz Biotechnology. Anti-GST (2624), anti-GFP (2956), anti-Tubulin (2146), anti-pEGFR pY1173 (2244), anti-EGFR (4267), anti-Shp2 pY580 (3754), anti-pErk1/2 (4370) and anti-Erk1/2 (9101) were from Cell Signalling Technology. Anti-RFP (A00682) was purchased from Genscript. Anti-6xHis (631210) was purchased from Takara. Anti-Shp2 antibodies were purchased from Santa Cruz Biotechnology, Cell Signal Technology, Sigma or Abcam. Anti-Grb2 pY160 was synthesised from Genscript[30]. Metafectin cell transfection reagents were purchased from Biontex.

**Western blots.** Cells were grown in 10-cm dishes, serum-starved overnight and stimulated with 10 ng/ml EGF for 15 minutes. Cells were lysed with buffer containing 50 mM Hepes, pH 7.5, 1% (vol/vol) igepal-C630, 1 mg/ml bacitracin, 1 mM ethylenediaminetetraacetic acid, 10 mM NaF, 1 mM sodium orthovanadate, 10% (vol/vol) glycerol, 50 mM NaCl, 1 mM phenylmethylsulfonyl fluoride and Protease Inhibitor Cocktail Set III (EMD Millipore). The detergent-soluble fraction was used for western blotting. Quantification of western blots was done using ImageJ software.

**Plasmids.** For the pull-down assay, gene fragments encoding Grb2 and Shp2 (28) were cloned into pGEX2T vector to construct GST-tagged Grb2 domains (GST-Grb2$_{NSH3}$, GST-Grb2$_{SH2}$ and GST-Grb2$_{CSH3}$) or Shp2 domains (GST-Shp2$_{WT}$, GST-Shp2$_{C459S}$, GST-Shp2$_{2SH2}$, GST-Shp2$_{PTP}$ and GST-Shp2$_{C69}$) using BamHI and NotI sites. For the cellular FLIM study, Grb2 genes (Grb2$_{WT}$ or Grb2$_{Y160E}$) were cloned in pECFP vector, and Shp2 C-terminal tail truncation (Shp2$_{\Delta69}$) was cloned in pcDNA-RFP vector using BamHI and NotI sites. For RFP-trap immunoprecipitation assay, Shp2$_{2SH2}$ and Shp2$_{PTP}$ were cloned in pcDNA-RFP vector using BamHI and NotI sites. For cell-based strep-tag pull-down experiments, Grb2 genes (Grb2$_{WT}$ or Grb2$_{Y160E}$) were amplified with an N-terminal strep-tag and cloned in a pcDNA6 vector using HindIII and PmeI sites. For the in vitro FRET assay, Shp2$_{\Delta69}$ was fused with an N-terminal BFP tag and a C-terminal GFP tag and was cloned in a pET28b vector. Mutagenesis (GST-Shp2$_{2SH2 R32A}$, GST-Shp2$_{2SH2 R138A}$, GST-Shp2$_{2SH2 R32/138A}$ and Grb2$_{R86A/Y160E}$) was carried out using Q5 site-directed mutagenesis kit (NEW England Biolabs) according to the manufacturer's protocol. To express recombinant proteins were in *Escherichia coli* for in vitro phosphatase assay, biophysical assay, pull-down assay and NMR studies. Genes of interest were amplified using standard PCR methods. Following designated restriction enzyme digestions, fragments were ligated into pET28b (Shp2$_{WT}$, Shp2$_{\Delta69}$, Shp2$_{2SH2}$, Shp2$_{2SH2 R32/138A}$, Shp2$_{NSH2}$, Shp2$_{CSH2}$, Shp2$_{PTP}$, Grb2$_{WT}$, Grb2$_{Y160E}$, Grb2$_{R86A/Y160E}$).

**Protein expression and purification.** Proteins were expressed in BL21 (DE3) cells. In all, 20 ml of cells grown overnight were used to inoculate 1 litre of LB media with antibiotic (50 μg/ml kanamycin for pET28b-backboned plasmids or 50 μg/ml ampicillin for pGEX4T1-backboned plasmids). The culture was grown at 37°C with

 COMMUNICATIONS BIOLOGY | https://doi.org/10.1038/s42003-021-01969-7

constant shaking (200 rpm) until the $OD_{600} = 0.7$. At this point, the culture was cooled down to 18°C and 0.5 mM of isopropyl β- d-1-thiogalactopyranoside was added to induce protein expression for 12 h before harvesting. Harvested cells were suspended in Talon buffer A (20 mM Tris, 150 mM NaCl and 1 mM β-ME, at pH 8.0) and lysed by sonication. Cell debris was removed by centrifugation (20,000 rpm at 4°C for 1 h). The soluble fraction was applied to an Akta Purifer System for protein purification. Elution was performed using Talon buffer B (20 mM Tris, 150 mM NaCl, 200 mM imidazole and 1 mM β-ME at pH 8.0). Proteins were concentrated to 2 ml and applied to a Superdex SD75 column using a HEPES buffer at pH 7.5 (20 mM HEPES, 150 mM NaCl and 1 mM TCEP, at pH 7.5).

**Pull-down/immunoprecipitation.** Purified protein or total transfected HEK293T cell lysate were prepared in 1 ml volume for pull-down or immunoprecipitation. In all, 50 μl of 50% glutathione beads (Sigma), Strep-Tactin beads, MERCK or RFP-Trap beads, Chromotek) were added and incubated for overnight. The beads were spun down at 5000 rpm for 3 minutes, supernatant was removed and the beads were washed with 1 ml lysis buffer. This washing procedure was repeated five times in order to remove non-specific binding. After the last wash, 50 μl of 2× Laemmli sample buffer were added, the samples were boiled and subjected to sodium dodecyl sulfate–polyacrylamide gel electrophoresis and western blot assay.

**Fluorescence lifetime imaging microscopy.** HEK293T cells were co-transfected with cyan fluorescent protein- and red fluorescent protein-tagged Grb2 and Shp2, respectively, (CFP-Grb2 and RFP-Shp2). After 24 h cells were seeded onto glass coverslips and allowed to grow for a further 24 h, fixed by the addition of 4% (w/vol) paraformaldehyde pH 8.0. Following 20 min incubation at room temperature cells were washed 6–7 times with phosphate-buffered saline (PBS), pH 8.0. Coverslips were mounted onto a slide with a mounting medium (0.1% p-phenylenediamine and 75% glycerol in PBS at pH 7.5–8.0). FLIM images were captured using a Leica SP5 II confocal microscope with an internal FLIM detector. CFP was excited at 860 nm with titanium–sapphire pumped laser (Mai Tai BB, Spectral Physics) with 710–920 nm tunability and 70 femtosecond pulse width. A Becker & Hickl SPC830 data and image acquisition card was used for time-correlated single-photon counting (TCSPC); electrical time resolution was 8 picoseconds with a pixel resolution of 256 × 256. Data processing and analysis were performed using a B&H SPC FLIM analysis software. The fluorescence decays were fitted to a single exponential decay model.

**Isothermal titration calorimetry.** ITC experiments were carried out using a MicroCal iTC200 instrument (Malvern), and data were analysed using ORIGIN7 software. To avoid heats associated with protein dissociation, $Grb2_{Y160E}$ was titrated into $Shp2_{\Delta69}$ at 25°C. The heat per injection was determined and subtracted from the control (buffer into $Shp2_{\Delta69}$) binding data. Data were analysed using a single independent site model using the Origin software.

**Microscale thermophoresis.** The Grb2 and Shp2 interactions were measured using the Monolith NT.115 MST instrument from Nanotemper Technologies. Proteins were fluorescently labelled with Atto 488 NHS ester (Sigma) according to the manufacturer's protocol. Labelling efficiency was determined to be 1:1 (protein to dye) by measuring the absorbance at 280 nm and 488 nm. A solution of unlabelled protein was serially diluted in the presence of 100 nM labelled protein. The samples were loaded into capillaries (Nanotemper Technologies). Measurements were performed at 25 °C in 20 mM HEPES buffer, pH 7.5, with 150 mM NaCl, 0.5 mM TCEP and 0.005% Tween 20, Data analyses were performed using Nanotemper Analysis software, v.1.5.41 and plotted using OriginPro 9.1.

**Biolayer interferometry.** BLI experiments were performed using a FortéBio Octet Red 384 using Anti-GST sensors. Assays were done in 384-well plates at 25°C. Association was measured by dipping sensors into solutions of analyte protein for 120 seconds and was followed by moving sensors to wash buffer for 120 seconds to monitor the dissociation process. Raw data show a rise in signal associated with binding followed by a diminished signal after application of wash buffer.

**In vitro phosphatase assay.** In vitro Shp2 activity assays were carried out using the Tyrosine Phosphatase Assay System (Promega) according to the manufacturer's manual. In brief, recombinant $Shp2_{\Delta69}$ was mixed with the phosphopeptide substrate in the presence or absence of different concentrations of Grb2 ($Grb2_{WT}$ or $Grb2_{Y160E}$). The method is based on measuring the absorbent change generated after the formation of a reaction mixture of molybdate:malachite green-phosphate complex of the free phosphate. The reaction time was 15 minutes for all experiments reported.

**MTT cell proliferation assay.** MDA468 proliferation assay was performed using Vybrant® MTT Cell Proliferation Assay Kit (ThermoFisher) according to the manufacturer's protocol. In brief, cells were transfected and seeded in 96-well plates. After overnight serum-starvation using phenol red-free medium, cells were labelled with 3-(4 5-dimethylthiazol-2-yl)-2 5-diphenyltetrazolium bromide (MTT) for 4 hours at 3°C and dimethyl sulfoxide (DMSO) was used as the solubilizing

agent to dissolve the formazan. Absorbance was measured at 470 nm. The percentage of viable cells was determined by normalising absorbance of culture medium treated with MTT and DMSO.

**Nuclear magnetic resonance spectroscopy.** Backbone resonances for the $^1H$, $^{15}N$, $^{13}C$ labelled sample of $Shp2_{NSH2}$ were assigned by recording a standard Bruker or BEST TROSY version[44] of 3D backbone resonance assignment spectra (HNCA, HNCOCA, HNCACB, CACBCONH, HNCO and HNCACO) with NUS sampling technique (25% fraction). The titration experiments of $Grb2_{SH2}$ into $Shp2_{NSH2}$ were recorded using $^{15}N$ HSQC pulse sequence from Bruker standard library. For that, samples of 200 μM uniformly $^{15}N$-labelled $Shp2_{NSH2}$ in 20 mM HEPES, 150 mM NaCl, 0.5 mM TCEP. pH 7.5 and 10% (v/v) D2O with 1:1, 1:2, 1:3, 1:4, 1:5, 1:6 molar ratio of $Shp2_{NSH2}$:$Grb2_{SH2}$ were prepared. All measurements were recorded at 25°C using a Brüker Avance III 750 MHz and 950 MHz spectrometers equipped with a Bruker TCI triple-resonance cryogenically coiled probes. Data were processed with NMRPipe[45] and analysed with CcpNmr Analysis software package[46]. CSPs for individual residues were calculated from the chemical shift for the backbone amide $^1H$ ($\Delta\omega_H$) and $^{15}N$ ($\Delta\omega_N$) using the following equation: $CSP = \sqrt{[\Delta\omega_H^2 + (0.154\,\Delta\omega_N^2)]}$ [47]. For the assignment of residues in the NSH2 domain of Shp2 coverage of non-proline residues was 81% complete. The first nine N-terminal residues remain unassigned and there is a gap in assignment of six and four residues at amino acid numbers 33–38 and 84–87, respectively.

**Statistics and reproducibility.** All data were expressed as mean and standard deviation. P values were determined by the Student's unpaired t test (Figs. 1e and 2c were calculated using Student's paired t test). The reproducibility was determined by using biological replicates and repeating the experiment two or three times as described in the figure legends.

**Reporting summary.** Further information on research design is available in the Nature Research Reporting Summary linked to this article.

## Data availability
The authors declare that all data supporting the findings of this study are available within the article and its supplementary information files. All data needed to evaluate the conclusions in the paper are present in the paper or the Supplementary Materials. The plasmids used in this study are available and require a material transfer agreement with University of Leeds.

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

## Acknowledgements

We thank Steve Homans for his assistance on the interpretation of NMR data and Amy Stainthorp (University of Leeds, School of Molecular and Cellular Biology) for critical discussion and useful comments. We acknowledge the Wellcome Trust (Grant 094232) and the University of Leeds for funding the NMR instrumentation. This work was funded by Cancer Research UK (Grant C57233/A22356).

## Author contributions

C.-C.L. and J.E.L. conceived of the project and designed experiments. C.-C.L., K.M.S. and Z.A. carried out experiments. L.W. and A.K. analysed NMR data. C.-C.L. and J.E.L. wrote the paper with input from all authors.

## Competing interests

The authors declare no competing interests.
