## [Peer Review File · Communications Biology]

Reviewers' comments:

Reviewer #1 (Remarks to the Author):

Lin et al., show that the adaptor protein Grb-2 binds to Shp-2 in its monomeric form, which is independent of the activation of growth factors. The authors did not only demonstrate their interaction but also identified the specific domains of Shp-2 and Grb-2 that interact with each other. Importantly, they additionally measured Shp-2 phosphatase activity within this context, demonstrating the functional relevance of this interaction. The manuscript is well written and the authors use adequate techniques to confirm their hypothesis. The concept is interesting, suggesting a plausible mechanism for growth of tumour cells independent of growth factors and activating mutations. However, I have some points and questions, which should be addressed.

- 1) The authors state that "proliferative or metastatic signalling could be driven by uncontrolled impact of mGrb2-mediated Shp2 activation". The authors base this statement upon their finding that ERK1/2 phosphorylation is affected by Grb2/Shp2 interaction. The manuscript would profit from a physiological assay, such as a proliferation assay of MDA-MB-468 cells, showing the impact of mGrb2/Shp2 interaction as well as the inhibitory peptide on tumour cell proliferation. It would strengthen their conclusion and increase the impact of the findings.
- 2) Why did the authors measure the binding of monomeric Grb2 and Shp2 Δ 69 intracellularly (Figure 1D), when these constructs did not interact in the BLI measurements? What is the reason behind this? And why did the authors not measure the interaction of monomeric Grb2 and all the different Shp2 constructs intracellularly to confirm the results depicted in Figure 1B?
- 3) Information regarding the generation of the different Shp2 plasmids is missing in the materials and method section. Please add this. In addition, further information on the Grb constructs should be added (how was the mutagenesis performed, or did the authors receive the plasmid from another source?)
- 4) Please add the molecular weight of the truncated Shp2 and Grb2 proteins, as it is very difficult to evaluate the western blots in figure 2C and 3B due to several bands. Larger sections of these blots should be shown in the figures. The current blot pictures are cropped and you can see that there are several protein bands on the blot, which have been cropped in half.
- 5) The manuscript lacks statistical analysis of the data. Please perform the analysis, where appropriate (i.e Figure 4B, C, J). Also, the manuscript lacks the number of experiments performed; please add the n's for all experiments in the figure legends.
- 6) The authors use the term Shp2FL in the text but Shp2WT in the figures, please use a consistent labelling of the constructs. Same with mGrb2 and Grb2Y160E.
- 7) Please perform quantification of the western blots in figure 1C, 2C, 3B and 4D (pERK) and subsequent statistical analysis.
- 8) Please add labellings to Figure 4B: Mark blue and red bars, as one cannot see which bars are mGrb2 and dimeric Grb2.
- 9) The blot next to the blots in Figure 4D is not mentioned in the figure legend.
- 10) References are missing for some statements. For instance line 59-61, 316-320, 320-321
- 11) Manuscript contains minor typing mistakes, for example line 276.

Reviewer #2 (Remarks to the Author):

In the manuscript "Grb2 binding induces phosphorylation-independent activation of the oncoprotein Shp2," Lin et al perform a series of biochemical experiments to characterize binding events between the protein tyrosine phosphatase (PTP) Shp2 and the adaptor protein Grb2.

Specifically, the authors show that a mutant of Grb2 (Y160E) that mimics the phosphorylated, monomeric form of the protein binds to both the SH2 domains and the PTP domain of Shp2. It is also convincingly demonstrated that Y160E Grb2 binding is sufficient to increase Shp2's PTP activity in a phosphorylation-independent manner.

Most of the manuscript's experiments are well carried out and the paper's biochemical data that detail domain-specific Grb2/Shp2 PTP interactions constitute a substantial contribution to the Shp2-regulation literature. I cannot however recommend the manuscript for publication in its current form. As detailed below, the portion of the paper that describes the authors' polypeptide inhibitor (PI) is not suitable for publication and needs to be either substantially revised or omitted.

Major Critiques

1. Structure and provenance of polypeptide inhibitor (PI): What is PI? What is its sequence? How did the authors design it and/or discover it? Puzzlingly, I could not find the answers to these fundamental questions in the manuscript.

2. Use of PI in Figure 4D: Did the authors experimentally confirm that the mysterious PI was successfully expressed from their vector in MDA-MD-468 cells? In a dose-dependent fashion? If so, how so? Regardless, the data in Figure 4D are not convincing. There is very little change in pERK1/2 levels in the no-EGF lanes regardless of the amount of vector added. It cannot be determined from visual inspection of the current data that a statistically significant, dosedependent reduction in pERK1/2 is caused by expression of PI.

3. Interpretation of PI data in Figure 4E: The authors claim that the data in 4E show that "binding of PI does not affect labelled Shp2NSH2 binding to Shp2PTP." In fact, if one compares the magenta data points (+ PI) and the cyan data points (no PI) in the binding curves, it can be seen that the addition of PI causes a rather substantial reduction in the apparent affinity of Shp2NSH2 for Shp2PTP. Strangely, the magenta line is drawn in such a way that it cannot possibly represent a best fit of the individual data points data. The line falls nowhere near the data points in the intermediate concentrations where the KD would lie (between roughly 800 and 1200 nM). Again, inspection of the individual data points reveals that PI has a rather large effect on the affinity of Shp2NSH2 for SHP2PTP, which is not consistent with the authors' model for how PI is exerting its effect on Shp2/Grb2 binding. (Also, if PI does not affect the Shp2NSH2/SHP2PTP interaction, why would the overall fluorescence change be so different in the presence of PI? See overall cyan $\Delta\Delta F$ cyan vs. magenta $\Delta\Delta F$.)

4. Inhibitory effect of PI (Figure 4J): The very mild inhibitory effect of 10 μ M PI on Shp2's PTP activity in the presence of mGrb2 raises more questions than it answers. Does PI inhibit Shp2 activity in the absence of mGrb2? Does it inhibit the isolated Shp2 catalytic domain's activity (SHP2PTP)? Does it attenuate the inhibitory activity of Shp2NSH2 on SHP2PTP? (One would expect so given the magenta/cyan binding data discussed above.)

Minor Critique

Lines 63-65: It is no longer true that "existing inhibitors targeting [Shp2] phosphatase activity show low selectivity." Small-molecule inhibitors that act by stabilizing Shp2's autoinhibited conformation are

extraordinarily selective. See Chen et al, *Nature*, 535, 148-152 and the many, many papers that have cited it.

We are extremely grateful for the comments we received from the Reviewers. We have made changes to the revised text in accordance with their recommendations and believe that the manuscript is greatly improved as a result.

Reviewer #1 (Remarks to the Author):

Lin et al., show that the adaptor protein Grb-2 binds to Shp-2 in its monomeric form, which is independent of the activation of growth factors. The authors did not only demonstrate their interaction but also identified the specific domains of Shp-2 and Grb-2 that interact with each other. Importantly, they additionally measured Shp-2 phosphatase activity within this context, demonstrating the functional relevance of this interaction. The manuscript is well written and the authors use adequate techniques to confirm their hypothesis. The concept is interesting, suggesting a plausible mechanism for growth of tumour cells independent of growth factors and activating mutations. However, I have some points and questions, which should be addressed.

1) The authors state that “proliferative or metastatic signalling could be driven by uncontrolled impact of mGrb2-mediated Shp2 activation”. The authors base this statement upon their finding that ERK1/2 phosphorylation is affected by Grb2/Shp2 interaction. The manuscript would profit from a physiological assay, such as a proliferation assay of MDA-MB-468 cells, showing the impact of mGrb2/Shp2 interaction as well as the inhibitory peptide on tumour cell proliferation. It would strengthen their conclusion and increase the impact of the findings.

We have conducted a proliferation assay with MDA468 cells transfected with Grb2_{WT} or Grb2_{Y160E}. The results of this experiment can be found in Figure 4E. The data show a significant increase in proliferation in the presence of the monomeric Grb2 (Grb2_{Y160E}), consistent with this form of the protein being able to release the autoinhibited conformation of Shp2 and elevate MAPK signalling. We did not include the effects of the inhibitory peptide in this new study since we have removed the entire study on the peptide from this manuscript in response to Reviewer #2's suggestion.

2) Why did the authors measure the binding of monomeric Grb2 and Shp2 Δ 69 intracellularly (Figure 1D), when these constructs did not interact in the BLI measurements? What is the reason behind this? And why did the authors not measure the interaction of monomeric Grb2 and all the different Shp2 constructs intracellularly to confirm the results depicted in Figure 1B?

In the BLI experiment (Figure 1B), we observed the significant interactions with monomeric Grb2 when we used Shp2_{WT} (the full length) and Shp2_{2SH2} (the tandem SH2 domains). PTP domain alone (Shp2_{PTP}) shows weak interaction whereas the C-terminal tail (Shp2_{C69}) does not interact with monomeric Grb2. We further used FLIM to confirm the interaction of Shp2 with monomeric Grb2 intracellularly. As it has been shown that the two tyrosines (Y542 and Y580) on Shp2 C-terminal tail can get phosphorylated and recruit Grb2 (via its SH2 domain). To remove the possibility of binding through this mode we used Shp2 construct in which the C-terminal tail which includes these tyrosines was deleted (RFP-Shp2 Δ 69).

In the revised version, we have addressed reviewer 1's comment on measuring the intracellular interaction of Shp2 domains with monomeric Grb2. We co-transfected CFP-monomeric Grb2 with RFP-Shp2_{2SH2} or RFP-Shp2_{PTP}. The subsequent immunoprecipitation experiment clearly shows the interactions with monomeric Grb2 are mediated by both Shp2

tandem SH2 domains and PTP domain (less pronounced, consistent with the *in vitro* studies.). This result is now included as Fig 1E.

3) Information regarding the generation of the different Shp2 plasmids is missing in the materials and method section. Please add this. In addition, further information on the Grb2 constructs should be added (how was the mutagenesis performed, or did the authors receive the plasmid from another source?)

We apologise for this error. We have added the requested information in the M&M section.

4) Please add the molecular weight of the truncated Shp2 and Grb2 proteins, as it is very difficult to evaluate the western blots in figure 2C and 3B due to several bands. Larger sections of these blots should be shown in the figures. The current blot pictures are cropped and you can see that there are several protein bands on the blot, which have been cropped in half.

We have added the molecular weight information of Shp2 and Grb2 proteins in the appropriate figures.

The original blot for Figure 2C contains the result of the pull down experiment using 'phosphorylated Grb2' to evaluate both the phosphotyrosine-independent and -dependent interactions. The phosphorylated Grb2 interacts with Shp2 SH2 domain through a canonical SH2-pY interaction.

On consideration, we decided that the phosphorylation-dependent interaction is irrelevant in this study therefore it has been removed. The original blot is shown here. We have replaced the figure with a repeated experiment using only the unphosphorylated Grb2. We have addressed the issue of the cropping of the blots by including more of the gels.

We apologise for the quality of Figure 3B due to the cross-reacting of antibodies, we have replaced this figure with a new repeat experiment.

5) The manuscript lacks statistical analysis of the data. Please perform the analysis, where appropriate (i.e Figure 4B, C, J). Also, the manuscript lacks the number of experiments performed; please add the n's for all experiments in the figure legends.

We have performed statistical analysis for Figures 1C, 1E, 2C, 3B, 4B, 4C, 4D, and 4E. The p values can be found included in the Figures.

We also added the 'n' value in the figure legends to indicate how many times we have performed the experiments.

6) The authors use the term Shp2FL in the text but Shp2WT in the figures, please use a consistent labelling of the constructs. Same with mGrb2 and Grb2Y160E.

We apologise for the mistake, we have corrected the errors. We used the name Shp2_{WT} through the manuscript. We also used Grb2_{Y160E} to correspond to with the use of the tyrosine phosphorylated mimic which is always monomeric. However, in cases where this mutant form was not directly used in an experiment, we have used mGrb2 to reflect the likely existence of a monomeric form (which is not the mutant). For example, in wild type cancer cells the Grb2_{Y160E} will not be present, but when Grb2 is at low expression levels mGrb2 will be present and capable of up-regulating signal transduction.

7) Please perform quantification of the western blots in figure 1C, 2C, 3B and 4D (pERK) and subsequent statistical analysis.

We have performed densitometric and statistical analysis for Figures 1C, 2C, 3B, and 4D (we also included pShp2).

8) Please add labellings to Figure 4B: Mark blue and red bars, as one cannot see which bars are mGrb2 and dimeric Grb2.

We apologise for the missing labels. These have now been added to the appropriate Figures.

9) The blot next to the blots in Figure 4D is not mentioned in the figure legend.

This figure has been removed in response to the suggestion from the reviewer 2.

10) References are missing for some statements. For instance line 59-61, 316-320, 320-321

We have added all of the requested references.

Due to the request of reviewer 2, we have removed the result of inhibitory peptide study therefore it is not necessary to add the references on lines 316-320 and 320-321.

11) Manuscript contains minor typing mistakes, for example line 276.

We have checked for typos and believe that these have been removed from the manuscript.

Reviewer #2 (Remarks to the Author):

In the manuscript "Grb2 binding induces phosphorylation-independent activation of the oncoprotein Shp2," Lin et al perform a series of biochemical experiments to characterize binding events between the protein tyrosine phosphatase (PTP) Shp2 and the adaptor protein Grb2.

Specifically, the authors show that a mutant of Grb2 (Y160E) that mimics the phosphorylated, monomeric form of the protein binds to both the SH2 domains and the PTP domain of Shp2. It is also convincingly demonstrated that Y160E Grb2 binding is sufficient

to increase Shp2's PTP activity in a phosphorylation-independent manner.

Most of the manuscript's experiments are well carried out and the paper's biochemical data that detail domain-specific Grb2/Shp2 PTP interactions constitute a substantial contribution to the Shp2-regulation literature. I cannot however recommend the manuscript for publication in its current form. As detailed below, the portion of the paper that describes the authors' polypeptide inhibitor (PI) is not suitable for publication and needs to be either substantially revised or omitted.

Major Critiques

1. Structure and provenance of polypeptide inhibitor (PI): What is PI? What is its sequence? How did the authors design it and/or discover it? Puzzlingly, I could not find the answers to these fundamental questions in the manuscript.

We have removed the study of PI.

2. Use of PI in Figure 4D: Did the authors experimentally confirm that the mysterious PI was successfully expressed from their vector in MDA-MD-468 cells? In a dose-dependent fashion? If so, how so? Regardless, the data in Figure 4D are not convincing. There is very little change in pERK1/2 levels in the no-EGF lanes regardless of the amount of vector added. It cannot be determined from visual inspection of the current data that a statistically significant, dose-dependent reduction in pERK1/2 is caused by expression of PI.

We do not have an antibody to confirm the expression of PI, however we have cloned a strep-tag fusion PI, which can be used for checking the expression. Since the current data is not sufficient to support our claim, we have removed the PI study.

3. Interpretation of PI data in Figure 4E: The authors claim that the data in 4E show that "binding of PI does not affect labelled Shp2NSH2 binding to Shp2PTP." In fact, if one compares the magenta data points (+ PI) and the cyan data points (no PI) in the binding curves, it can be seen that the addition of PI causes a rather substantial reduction in the apparent affinity of Shp2NSH2 for Shp2PTP.

The ΔF is different but the affinities are still close.

Strangely, the magenta line is drawn in such a way that it cannot possibly represent a best fit of the individual data points data.

We thank the Reviewer for pointing out this mistake. We re-examined the data and we found an error in plotting of the Figures using the Origin software. The correct fitting curve from the MST analysis software is shown below; fortunately, this does not change the result.

The line falls nowhere near the data points in the intermediate concentrations where the KD would lie (between roughly 800 and 1200 nM). Again, inspection of the individual data points reveals that PI has a rather large effect on the affinity of Shp2NSH2 for SHP2PTP, which is not consistent with the authors' model for how PI is exerting its effect on Shp2/Grb2 binding.

PI does have a large effect on the affinity of Shp2_{NSH2} for SHP2_{PTP}, it binds to Shp2_{NSH2} therefore Grb2 can not bind to Shp2_{NSH2}

(Also, if PI does not affect the Shp2NSH2/SHP2PTP interaction, why would the overall fluorescence change be so different in the presence of PI? See overall cyan $\Delta\Delta F$ cyan vs. magenta $\Delta\Delta F$.)

The MST technique in this case is measuring the movement of labelled protein. Here, the labelled Shp2_{NSH2} is bound with excess of PI (magenta), therefore its movement (reflected in

$\Delta\Delta F$) is different from the Shp2SH2 (cyan) only due to the change of size, charge, hydration shell or conformation. However, this difference in $\Delta\Delta F$ does not affect the binding affinity.

4. Inhibitory effect of PI (Figure 4J): The very mild inhibitory effect of 10 μM PI on Shp2's PTP activity in the presence of mGrb2 raises more questions than it answers. Does PI inhibit Shp2 activity in the absence of mGrb2? Does it inhibit the isolated Shp2 catalytic domain's activity (SHP2PTP)? Does it attenuate the inhibitory activity of Shp2NSH2 on SHP2PTP? (One would expect so given the magenta/cyan binding data discussed above.)

We are grateful for the thoughtful questions, we have not pursued these because we have removed the PI data. However, should we decide to carry on the PI study in the future, we would definitely take these comments into account.

Minor Critique

Lines 63-65: It is no longer true that "existing inhibitors targeting [Shp2] phosphatase activity show low selectivity." Small-molecule inhibitors that act by stabilizing Shp2's autoinhibited conformation are extraordinarily selective. See Chen et al, Nature, 535, 148-152 and the many, many papers that have cited it.

We have edited according to these comments and included the appropriate references.

REVIEWERS' COMMENTS:

Reviewer #1 (Remarks to the Author):

The manuscript from Lin et al., has improved through the revision and is now clearer and contains sufficient experiments to confirm their hypothesis. The authors have made an important contribution to the clarification of the regulation of cellular signalling through Shp-2/Grb-2 interaction and activation. I only have a few minor points, which should be corrected.

1) The materials and method section is missing a section explaining the statistics (which tests were used and if graphs show SEM or SD).

2) The authors show that mGrb2 is phosphorylated under non-stimulated conditions. I feel, the manuscript would profit from a short explanation in the discussion of how the authors hypothesize that Grb2 is phosphorylated in absence of a stimulation (growth factor).

3) Figures 4C and 4E contains data from only two independent experiments. Performing statistics on only two values is not plausible. Were there perhaps several samples within every experiment (i.e. triplicates etc.)? If so, the authors should state this in the figure legend.

Reviewer #2 (Remarks to the Author):

In the revised manuscript "Grb2 binding induces phosphorylation-independent activation of Shp2," Lin et al present a series of biochemical experiments to characterize binding events between the protein tyrosine phosphatase (PTP) Shp2 and the adaptor protein Grb2.

The authors have satisfactorily responded to my comments on the original manuscript, and I am now happy to recommend the revised version for publication pending one small revision.

Minor Critique:

Although the authors omitted the polypeptide inhibitor (PI) experiments in response to my earlier comments, it appears that they forgot to remove PI from Table 1. The entries in Table 1 that still refer to PI should be removed before publication.

Reviewer #1 (Remarks to the Author):

The manuscript from Lin et al., has improved through the revision and is now clearer and contains sufficient experiments to confirm their hypothesis. The authors have made an important contribution to the clarification of the regulation of cellular signalling through Shp-2/Grb-2 interaction and activation. I only have a few minor points, which should be corrected.

1) The materials and method section is missing a section explaining the statistics (which tests were used and if graphs show SEM or SD).

We have added the information regarding the statistics in the Methods. We also indicated the statistics information in the figure legends.

2) The authors show that mGrb2 is phosphorylated under non-stimulated conditions. I feel, the manuscript would profit from a short explanation in the discussion of how the authors hypothesize that Grb2 is phosphorylated in absence of a stimulation (growth factor).

We appreciate the suggestion, we have added the requested information in the discussion together with proper citations (page 13).

3) Figures 4C and 4E contains data from only two independent experiments. Performing statistics on only two values is not plausible. Were there perhaps several samples within every experiment (i.e. triplicates etc.)? If so, the authors should state this in the figure legend.

They are done with multiple samples within every experiment, we have added the information in the figure legends.

Reviewer #2 (Remarks to the Author):

In the revised manuscript "Grb2 binding induces phosphorylation-independent activation of Shp2," Lin et al present a series of biochemical experiments to characterize binding events between the protein tyrosine phosphatase (PTP) Shp2 and the adaptor protein Grb2.

The authors have satisfactorily responded to my comments on the original manuscript, and I am now happy to recommend the revised version for publication pending one small revision.

Minor Critique:

Although the authors omitted the polypeptide inhibitor (PI) experiments in response to my earlier comments, it appears that they forgot to remove PI from Table 1. The entries in Table 1 that still refer to PI should be removed before publication.

We thank reviewer 2 for pointing this out, we apologise for the mistake and we have removed the extra table.